# Variation Tendency of Coastline under Natural and Anthropogenic Disturbance around the Abandoned Yellow River Delta in 1984–2019

Zhipeng Sun 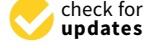 and Xiaojing Niu *

State Key Laboratory of Hydroscience and Engineering, Department of Hydraulic Engineering, Tsinghua University, Beijing 100084, China; szp20@mails.tsinghua.edu.cn
* Correspondence: nxj@tsinghua.edu.cn; Tel.: +86-1381-073-5396

**Abstract:** The coast around the Abandoned Yellow River Delta underwent significant changes under anthropogenic disturbance. This study aims to reveal the variation of the coastline, tidal flat area, and intertidal zone slope before, during, and after extensive reclamation during the period of 1984–2019 using satellite remote sensing images. In order to eliminate the influence of the varying water level, a new coastline correction algorithm had been proposed under the condition of insufficient accurate slope and water level data. The influence of seawalls on slope estimation were considered in it. The spatiotemporal evolution of coast had been analyzed and confirmed to be reasonable by comparing with the observed data. The results show that the coast can be roughly divided into a north erosion part and a south deposition part. Affected by reclamation, their tidal flat area in 2019 is reduced to only 43 and 27% of original area in 1984, respectively, which results in a continuous decrease in the tidal flat width. The adjustment of the tidal flat profile makes the slopes steeper in the erosion part, while the slopes in the deposition part remain stable. The reclamation has stimulated a cumulative effect as the disappearance of the intertidal zone, which may lead to the destruction of biological habitats.

**Keywords:** Abandoned Yellow River Delta; remote sensing images; spatiotemporal changes; correction algorithm; coastline; tidal flat

## 1. Introduction

Tidal flat is a transition zone between marine and terrestrial ecosystems and contains abundant natural resources. It can not only provide biological habitats, buffer the impact of extreme climates, and maintain the diversity and stability of the ecosystem, but it can also create an important marine economic development zone that integrates port trade, landscape tourism, sea salt production, and other functions; therefore, it has great development potential [1–3]. The development value of tidal flat resources has laid the foundation for the rapid growth of population and social economy in recent decades. However, the rapid industrialization, urbanization, and the extensive anthropogenic disturbances have severely influenced the changes of coastlines and the reduction in tidal flat resources, which bring a series of environmental and ecological problems, such as habitat destruction, soil degradation, biodiversity loss, etc. [4–7]. Therefore, study on the spatiotemporal changes of coastline and tidal flat resources is of strategic significance for future coastal cities to keep the balance of the development and protection of tidal flat resources, alleviate the substantial conflicts between economic development and environmental protection, and achieve sustainable development [8–11].

The Yellow River, known as "the cradle of Chinese civilization", has historically been recognized for the high suspended sediment load that inspired its name [12]. Therefore, a large amount of sediment is easily carried by the Yellow River and transported to the estuary, thereby the fine-grained sediment deposits form a delta [13]. Throughout history,

the downstream of the Yellow River in the North China Plain has been diverted many times. The Abandoned Yellow River Delta, as one of the most typical modern abandoned deltas in the world, was formed between 1128 and 1855, when the Yellow River flew into the Yellow Sea from Jiangsu Province [14]. After 1855, the Yellow River debouched into the Bohai Sea, thus the Abandoned Yellow River Delta lost its main sediment source and, consequently, began to erode [15]. Since the Ancient Yellow River carried a large amount of sediment and deposited around the Abandoned Yellow River Delta, Jiangsu Province, where the Abandoned Yellow River Delta, is located has about one quarter of the total tidal flat area of China. The abundant tidal flat resources become a unique advantage for the development of Jiangsu Province. Since China's reform and opening up in the late 1970s, extensive anthropogenic reclamation has been conducted and resulted in a rapid reduction in the natural tidal flat area. In the last decade, the local policy has shifted to strict protection of the coastline, e.g., strict control of reclamation. It is of interest to reveal the variation tendency of the coastline, tidal flat area, and intertidal zone slope before, during, and after extensive reclamation, to show the influence of the policy for the utilization and conservation of the coastal zone in China over the past several decades.

Based on satellite remote sensing images to extract and analyze the coastlines, there have been a lot of studies in recent years [16–22]. For example, Luijendijk et al. [22] successfully made a global-scale analysis of the occurrence of sandy beaches and rates of shoreline change therein using the satellite-derived shorelines (SDS). Their method is similar to that of Hagenaars et al. [23], by using the moving average composite images to decrease errors from clouds, waves, sensor corrections, georeferencing, and improve the accuracy of SDS to subpixel precision. The method is of high efficiency to obtain the satellite-derived shorelines. It should be noted that influenced by tides and waves, the instantaneous water edges are always in dynamic change at different temporal and spatial scales [24]. Especially for the coast with a relatively gentle slope or a large tidal range, the instantaneous water edge changes dramatically and the method based on moving average composite images needs more images to eliminate the effect of the varying water level. Castelle et al. [25] pointed out that publicly available satellite derived data are associated with great uncertainty for beaches experiencing a large tidal range (>2 m). In order to better describe the coastline changes in the case of gentle slope and large water level variation, two types of coastline indicators are commonly used [26], one is based on a visually discernible coastal feature such as the high tide line [27], and the other one is datum-based shoreline indicators [28,29]. Although the high tide lines are not easily affected by sea level fluctuations compared with the instantaneous shorelines, Crowell et al. [30] pointed out that the actual high tide lines were not obvious. Therefore, the mean sea level tidal-datum-based coastline is widely adopted by researchers and also considered to evaluate the changes of coastline around the Abandoned Yellow River Delta in this study. Actually, accurate intertidal zone slope and water level data are key to obtain the datum-based coastline.

However, due to the relative lack of measured slope data in most areas of China, in many studies of coastline change, the slope data required for coastline correction is usually calculated by two or three satellite images, and the endpoint rate is used to reflect the coastline change [28]. However, the endpoint rate is easily affected by the error of the shoreline position in the image; therefore, there is a large error between the endpoint rate and the actual coastline change rate. For this reason, in order to improve the reliability of the analysis results, statistical analysis based on a large number of instantaneous water edge positions is used to study the coastline migration rate [31]. However, without eliminating the influence of the varying water level, the statistical analysis is difficult to observe the changes of the coastline in a short period of time. In addition, even if the intertidal zone slope estimation and tidal level correction are carried out, the results still have a large deviation from the actual coastlines without considering the change of the seawall or temporary dike.

In order to better analyze the spatial and temporal changes of the coastline around the Abandoned Yellow River Delta in 1984–2019, this study used Landsat series satellite remote sensing images to extract the instantaneous water edges as the data basis. At the same time, the new coastline correction algorithm was proposed based on the intertidal zone slope estimation and water level correction, and the natural and anthropogenic disturbance were further considered in it. The algorithm can not only effectively achieve the coastline correction, thereby reflecting the spatiotemporal changes of the coastline, but also is generally applicable to areas without accurate data of water level and slope. The study results have strategic significance for revealing the past development problems around the Abandoned Yellow River Delta and achieving future sustainable economic development planning and coastal ecological protection.

## 2. Study Area and Data Source

### 2.1. Study Area

The study area is around the abandoned Yellow River delta from the Shaoxiang Estuary to Dongtai Estuary, as shown in Figure 1, which is the middle part of the coast in Jiangsu Province and located to the west of the Yellow Sea. This part of the coast in Jiangsu Province is silt coast, and generally has a quite gentle intertidal zone slope. The cause of the tidal flat is closely related to the Ancient Yellow River, and it is greatly affected by the diversion of the Ancient Yellow River. The variation tendency of the coastline in this area was dramatic under both natural and anthropogenic disturbance. Since China's reform and opening up, the Jiangsu policies have shifted from extensive reclamation to strict protection, and this area underwent an extensive reclamation that has had a huge influence on the changes of the coastline.

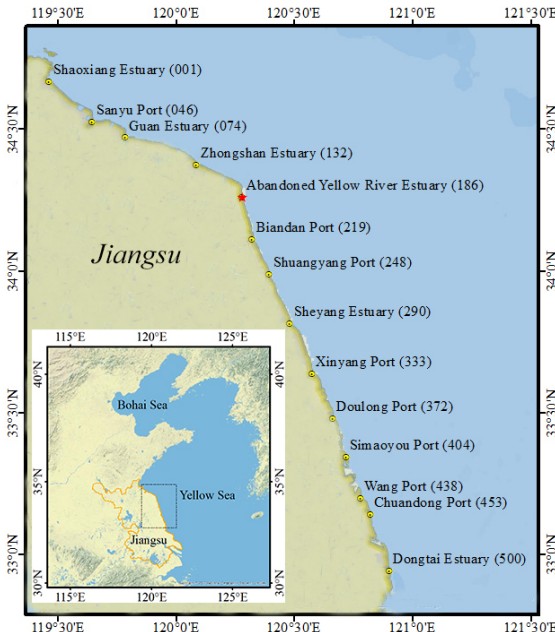

**Figure 1.** Study area and division of coast transect. The number in brackets is the transect number for main estuaries.

It is hard to obtain long-term observed data on the hydrodynamics of this area. From the literatures and the public ocean tidal model, it can be known that the tidal range is relatively large in the study area and varying from the north part to the south part. The maximum possible tidal range is 5–7 m off the southern coast, while the maximum possible tidal range off the northern coast is about 4 m [32]. The spatial variation of tidal range cannot be ignored in such a large area. The slope in this area is about 1/1000 to

1/10,000. Combining with the gentle slope and large tidal range, it should be noted that the instantaneous water edges in this area are greatly affected by the varying water level.

## 2.2. Coastal Baseline Selection and Transect Division

The coastal baseline selection and transect division are the basis for obtaining instantaneous water edges and seawall positions, and further analyzing the spatial and temporal changes of the coastline. The coastline baselines are determined based on the principle that they are approximately parallel to the coast direction and placed in the sea side. Based on the coastline baselines, the relative baseline distance from the instantaneous water edges to the coastline baselines can be determined. The positive direction points to the sea side. The relative baseline distance from the land side is negative, while from the sea side it is positive. The changes of the relative baseline distance reflect the coastline changes. The coastline baselines are determined based on the principle that they are approximately parallel to the coast direction. In order to better fit the coast direction, the segmented baselines are adopted. According to the tortuous characteristic of the coast around the Abandoned Yellow River Delta, a total of seven coastline baseline segments are set, as shown in Table 1. In order to reflect the spatiotemporal changes of the coastline, transects with the equal interval of 500 m are arranged along the baseline at the location of each coast transect as much as possible. In total, 500 transects are divided around the Abandoned Yellow River Delta from Shaoxiang Estuary, denoted by number 001, to Dongtai Estuary, denoted by number 500.

**Table 1.** Coastline baselines basic settings.

| Path | Row | Baseline | Transect | Initial Coordinate Position | Final Coordinate Position | Length (m) |
|------|-----|----------|----------|------------------------------|----------------------------|------------|
| 120 | 36 | 1 | 001–077 | 119.501° E, 34.710° N | 119.851° E, 34.526° N | 38,071 |
| | | 2 | 077–125 | 119.851° E, 34.526° N | 120.090° E, 34.437° N | 24,082 |
| | | 3 | 125–171 | 120.090° E, 34.437° N | 120.307° E, 34.335° N | 22,962 |
| | | 4 | 171–219 | 120.307° E, 34.335° N | 120.371° E, 34.124° N | 24,169 |
| | | 5 | 219–233 | 120.371° E, 34.124° N | 120.404° E, 34.065° N | 7215 |
| 119 | 37 | 6 | 233–363 | 120.404° E, 34.065° N | 120.696° E, 33.532° N | 64,781 |
| | | 7 | 363–500 | 120.696° E, 33.532° N | 120.973° E, 32.963° N | 71,805 |

## 2.3. Satellite Remote Sensing Image Data

As an important way to obtain geographic information, remote sensing technology has the advantages of a relatively low data acquisition cost, wide coverage, and high spatial resolution; thus, it has been widely used in shoreline extraction for a long time. Pardo-Pascual et al. [33] believed that the Landsat images' detection accuracy of instantaneous water edges was comparable to high-resolution technology; therefore, the extraction of instantaneous water edges from Landsat series satellite images could relatively accurately describe landform features. In this study, the cloud density and the image clarity were used as the standard for satellite images screening. On this basis, a total of 501 satellite images, including Landsat5 TM(L5), Landsat7 ETM+(L7), and Landsat8 OLI_TIRS(L8), were obtained around the Abandoned Yellow River Delta from 1984 to 2019 through the Google Earth Engine platform. All the selected satellite images were used to extract the instantaneous water edges and seawalls.

Due to the large span of the study area, two satellite images were required to achieve full coverage. A total of 274 satellite images were screened out with path and row number (120, 36), of which the number of satellite images from L5, L7, and L8 were 134, 105, and 35, respectively. A total of 227 satellite images were screened out with path and row number (119, 37), of which the number of satellite images from L5, L7, and L8 were 112, 86, and 29, respectively. The details of all the satellite remote sensing image data are shown in Figure 2.

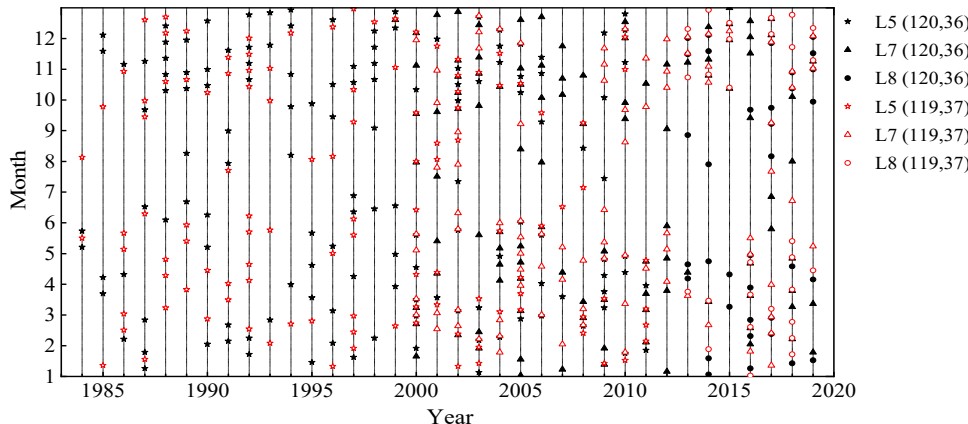

**Figure 2.** Satellite remote sensing image data.

*2.4. Validation Data*

In order to verify the rationality and accuracy of the results from the remote sensing analysis, a series of the coastline change rate data from historical observation is used as validation data. From 1980 to 1997, the Jiangsu Tidal Flat Management Bureau and Jiangsu Agricultural Resources Development Bureau conducted 18 years of continuous observations on the changes of the coastline in Jiangsu. They arranged a set of transects perpendicular to the local seawall on the representative coast. Along each transect, a number of cement piles were set from the seawall to the lowest tidal level as far as possible. Every year the elevations of the top of piles were measured, and the height of each pile above the surface of tidal flat was measured every season, as to obtain the profile of tidal flat at each transect. The data on the annual migration rate of average high tide level are available. The data can be found in the report "Research on Comprehensive Development Strategy of Jiangsu Coastal Area. Tidal Flat Volume. Evaluation and Reasonable Development and Utilization of Tidal Flat Resources in Jiangsu Coastal Area" [32].

**3. Methodology**

The semi-automated shoreline extraction method proposed by Daniels [34] is used to extract instantaneous water edges from satellite images as the basic data, while a visual interpretation is more convenient to extract the positions of seawalls or temporary dikes. Due to the fact the slope is very gentle around the Abandoned Yellow River Delta, the instantaneous water edge is greatly affected by the water level at the moment that the satellite image was taken. Without eliminating the influence of the fluctuating water level, it is nearly impossible to reliably analyze the actual spatiotemporal changes of the coastline using the instantaneous water edges. Thus, a correction to eliminate the influence of the fluctuating water level is first conducted, in order to obtain the datum-based coastline from the instantaneous water edge. There are the following two difficulties: (1) the slope in this area is varying spatiotemporally, and no available data can be used for each transect; (2) although we can obtain the tidal level from some public databases, it is not possible to obtain the accurate local water level at every moment. Moreover, the public tidal data are known to be not very accurate in the nearshore shallow water area, which has been confirmed by comparing them with the limited available measure data. To solve these problems, a new coastline correction algorithm that contains an intertidal zone slope estimation and water level correction is used to obtain the datum-based coastlines.

*3.1. Coastline Correction Algorithm Process*

Affected by the varying water level, the long-term variation trend of the coastline is concealed under the drastic change of instantaneous water edges. The distance of the instantaneous water edge from the baseline (denoted by $C_{IWE}$) at transect number $s$ at time $t$ can be divided into two parts, as expressed in Equation (1), including the distance of the

mean sea level datum-based coastline (denoted by $C_{MSL}(s,t)$) and a deviation due to the fluctuation of water level ($C_{WLF}$).

$$C_{IWE}(s,t) = C_{MSL}(s,t) + C_{WLF}(s,t) \tag{1}$$

The object is to determine $C_{MSL}$, which is also a function of location and time. As $C_{IWE}$ is known after image data extraction, the key is to estimate $C_{WLF}$. The variation of the water level can be divided into several components, including the long-term variation of the mean sea level, daily variation due to the tide, and the higher frequency variation corresponding to surges and wind waves. Assuming the tidal flat profile at each transect is approximately a slope, basically $C_{WLF}$ can be calculated using Equation (2).

$$C_{WLF}(s,t) = \frac{H_{total}(s,t)}{S(s,t)} = \frac{H_T(s,t) + H_M(t) + H_R(s,t)}{S(s,t)} \tag{2}$$

In which, $H_{total}(s,t)$ is the local water level at transect number $s$ and time $t$. $S(s,t)$ is the intertidal zone slope. $H_{total}(s,t)$ is further divided into three components. $H_T(s,t)$ is the tidal level, and is varying with time as well as locations, because the tidal range within the study area varies largely. The tidal level can be obtained directly from the public ocean tidal model [35], which includes the main shore-period tidal constituents. $H_M(t)$ is the mean sea level changes, which is treated as a constant within the area and only varying with time. The monthly mean sea level is approximately adopted, which is obtained from the "2019 China Sea Level Bulletin" [36]. $H_R(s,t)$ is called the random fluctuation of the water level, contributed to by surges and waves, as well as errors in the estimation of tidal level.

However, the random fluctuation of the water level and intertidal zone slope are unknown. An estimation for the two variables should be given firstly. Then, the procedure is divided into the following three major steps:

(1)　to estimate the random fluctuation of water level $H_R(s,t)$;
(2)　to obtain the intertidal zone slope at each transaction and time $S(s,t)$;
(3)　to calculate the mean sea level datum-based coastline (MSL coastline) $C_{MSL}(s,t)$.

The algorithm is designed based on the predictor–corrector technique. The detailed procedure of the coastline correction algorithm proposed in this study is shown in Figure 3, each step will be introduced in the following.

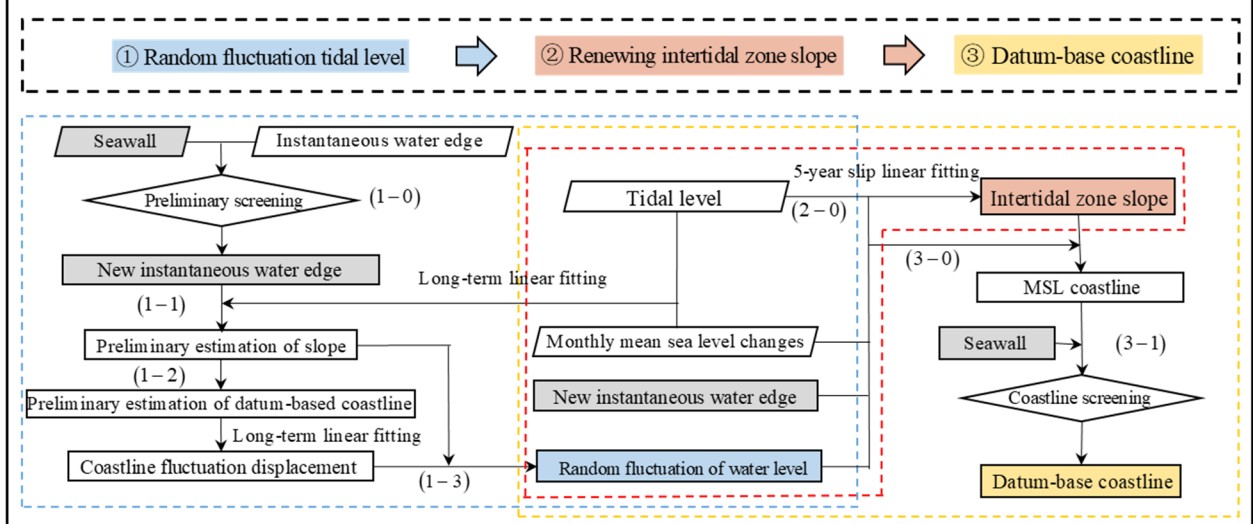

**Figure 3.** Procedure of the coastline correction algorithm.

*3.2. Random Fluctuation of Water Level*

The random fluctuation of water level is also reflected in the instantaneous water edge data, this section aims to decouple it from the variation of water edges. It includes several sub-steps, which are shown in the blue box of Figure 3.

Firstly, a preliminary estimation of the time averaged slope at each transect is given in sub-step (1–1). Generally, the instantaneous water edge approaches the coast at high tidal level and recedes to the sea at low tidal level, which provides the basis for the estimation of the intertidal zone slope. The influence of mean sea level $H_M$ and tidal level $H_T$ on the changes of the instantaneous water edge $C_{IWE}$ is the most significant. Although the tidal level obtained from the public ocean tidal model may have phase lag and deviation in tidal range in shallow water area, the major tidal characteristics can be well given. Therefore, the gradient obtained from the linear regression between $C_{IWE}$ and $H_M + H_T$ can be used for the preliminary estimation of intertidal zone slope.

Using linear regression is based on the assumption that the profile of tidal flat is a slope, which can be a reasonable approximation for the tidal flat in a natural environment. However, when the seawalls are located in the intertidal zone, the instantaneous water edges touch the seawalls at middle or high tidal level, and the slope directly estimated using the previous calculation method would introduce huge errors. Therefore, before the estimation of slope, transects at each image are checked and marked whether the positions of the seawall and instantaneous water edge are overlapping. The sub-step is called data screening and numbered as (1–0). If a transect is marked "overlap", it would not be considered in the linear regression for estimating the slope, in order to minimize the influence of anthropogenic disturbance. Considering the resolution of image, the distance between the instantaneous water edge and the seawall less than 60 m is used as the critical value for data screening.

Secondly, a preliminary correction of coastline is obtained considering $H_M + H_T$ as the first approximation of the water level, in order to roughly eliminate the variation of datum-based coastline. The preliminary estimation of datum-based coastline is calculated using Equation (3) in sub-step (1–2).

$$C'_{MSL} = C_{IWE} - \frac{H_T + H_M}{\overline{S_0}} \tag{3}$$

where $C'_{MSL}$ is the preliminary estimation of the datum-based coastline, and $\overline{S_0}$ is the preliminary estimations of intertidal zone slope after the spatial slip average calculation. To weaken the influence of errors in the preliminary estimation of slope, a spatial average among the adjacent transects within a range of 2 km is adopted to obtain $S_0(s)$.

Thirdly, the random fluctuation of water level is estimated as sub-step (1–3). The calculation is based on Equation (2), but because we only have the preliminary slope and preliminary estimation of datum-based coastline, the formula is expressed as follows:

$$H_g = \left(C_{IWE} - \overline{C'_{MSL}}\right)\overline{S_0} - (H_T + H_M) \tag{4}$$

$\overline{C'_{IWE}}$ is the preliminary estimation of datum-based coastline after long-term linear fitting, to weaken the influence of errors induced in previous sub-steps. Considering the major part in the random fluctuation of water level has local spatial consistency, a spatial average among the adjacent transects within a range of 2 km is adopted to obtain the random fluctuation of water level for further analysis. Then, an estimation of the water level at each transect $s$ and time $t$ is given as follows:

$$H_{total}(s, t) = H_T + H_M + \overline{H_g} \tag{5}$$

### 3.3. Renewing Intertidal Zone Slope and Datum-Based Coastline

The core of the second step is to improve the accuracy of intertidal zone slope, which is shown in the red box in the middle of Figure 3. After the new estimation of water level is obtained, the slope at each transect can be renewed, through the relationship between the total water level and instantaneous water edges.

It should be noted that the slope of each transect changes with time. Theoretically, we can obtain the time-averaged slope between the taken moments of two images, when the instantaneous water edge and corresponding water level are accurate enough. However, it is hard to be absolutely accurate. Considering the errors induced by image resolution and others, a sliding window fitting is adopted to obtain a multi-year-average slope in sub-step (2–0). In this study, the sliding window is set to be 5 years corresponding to China's five-year plan.

Generally, the intertidal zone slopes will not change significantly within a certain spatial range, in order to further improve the accuracy of the intertidal zone slopes of each coastal transect, the 2-kilometer spatial scale is used for the spatial slip average calculation of the intertidal zone slopes.

Then, using the renewing slope and the estimation of water level, the datum-based coastline $C_{MSL}$ can be calculated using Equations (1) and (2) in sub-step (3–0).

Considering the influence of anthropogenic disturbance on the MSL coastlines, it is necessary to compare the MSL coastlines with the seawalls. When the seawalls overlap the MSL coastlines, the seawalls are taken as datum-based coastlines, otherwise, the MSL coastlines are taken as the datum-based coastlines. The coastline screening is established as sub-step (3–1).

In addition, the instantaneous water edges and seawalls always overlap in the satellite images of certain coastal transects in some years, resulting in the lack of instantaneous water edge data for that year at preliminary screening. For this phenomenon, although the coastal transects with missing data are impossible to conduct post-analysis, in terms of probability, it is quite certain that the MSL coastlines overlaps with the seawalls. Therefore, the seawalls can be directly used as the datum-based coastlines in this type of the coastal transects.

### 3.4. Validity Analysis of Coastline Correction Algorithm

By establishing a relationship between the coastlines and time, it can reflect the temporal changes of coastline position. Figure 4 shows the temporal changes of coastline and instantaneous water edges at the location of some representative coastal transects around the Abandoned Yellow River Delta. Before the correction, the distributions of instantaneous water edge are relatively discrete; therefore, it is difficult to judge the changes of the coastline. However, after processing by the coastline correction algorithm proposed in this study, the changes of coastline can be clearly seen.

It can be seen that the present coastline correction algorithm can perform well in areas with large tidal range variation. Compared with the method based on the moving average composite images [22,23], both have their own advantages. The method based on the moving average composite images is not limited by the number of images used and has a high degree of automation based on the platform of the Google Earth Engine; therefore, it is more efficient to analyze the variation tendency of coastline. This type of method has been extended to the analysis of global coastlines, but the accuracy of the analysis results for specific areas with a high tidal range or gentle slope needs to be further verified. The present algorithm is a little complicated but has good applicability for coastline analysis in areas without accurate data of water level and slope.

In the analysis process of coastline change, the whole time series least-squares fitting is used to analyze the variation tendency of coastline in many studies. Actually, most of the coastal transects around the Abandoned Yellow River Delta often show staged changes, as shown in Figure 5, instead of linear changes, as shown in Figure 4. For this reason, when analyzing the spatiotemporal changes of coastline later, the process of coastline change analysis has considered the different development stages.

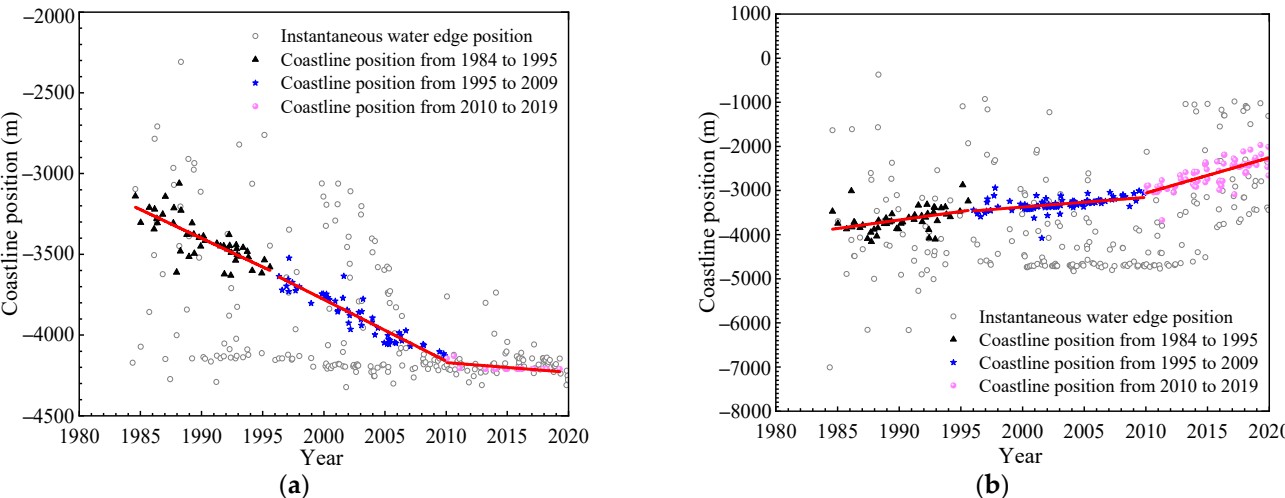

**Figure 4.** (**a**) The temporal changes of coastline position at the location of transect 81; (**b**) The temporal changes of coastline position at the location of transect 229; (**c**) The temporal changes of coastline position at the location of transect 331; (**d**) The temporal changes of coastline position at the location of transect 431.

**Figure 5.** (**a**) The temporal changes of coastline position at the location of transect 235; (**b**) The temporal changes of coastline position at the location of transect 292.

## 4. Results and Discussion

### 4.1. Phased Anthropogenic Reclamation

In order to obtain the overall variation tendency of a tidal flat resource under natural and anthropogenic disturbance around the Abandoned Yellow River Delta in 1984–2019, the natural tidal flat area, the reclamation area, and the total coastal area are defined and calculated. The natural tidal flat area in this study refers to the area between the seawall and the datum-based coastline, and the area between the baseline and the seawall (or temporary dike) is defined as the reclamation area. The area baseline is defined as the initial seawall in 1984. The total coastal area is the summation of the natural tidal flat area and the reclamation area, which is the area between the area baseline and the datum-based coastline. The changes of the area between the baseline and seawall, the datum-based coastline, over time, reflects the changes of the tidal flat resource under natural and anthropogenic disturbance.

As the policy for the utilization and conservation of the coastal zone in China has been changing over the past several decades, the coast around the abandoned Yellow River underwent a period of extensive reclamation, which can be taken as a typical coast with significant changes under anthropogenic disturbance. In order to reveal the specific changes of anthropogenic reclamation, the reclamation area in each year is calculated by accumulating all the areas at each transect, which are the product of the longitudinal distance between the seawall and the area baseline at each transect and the lateral distance between adjacent transects. The changes of the anthropogenic reclamation area around the Abandoned Yellow River Delta in 1984–2019 are shown in Figure 6. According to the change of the anthropogenic reclamation area in Figure 6, it is easily found that the net increase in the anthropogenic reclamation area is relatively slow before 1995 and after 2010, while relatively fast from 1995 to 2010. After China' reform and opening up, Jiangsu Province formulated a series of coastal tidal flat development strategies to guide people to strengthen the development and utilization of tidal flat resources, in order to promote the development of the coastal economy. Before 1995 or the end of the "Eighth Five-Year Plan" period, the development and utilization of tidal flat resources in Jiangsu entered a period with the main purpose of pursuing economic benefits. From the "Ninth Five-Year Plan" to the "Eleventh Five-Year Plan" period (1996–2010), a series of large-scale tidal flat development projects were initiated, which left an extensive reclamation period in history. After 2010, Jiangsu Province implemented the "Outline of Jiangsu Coastal Flat Reclamation Development and Utilization Plan" and emphasized the sustainable development of tidal flat resources. Therefore, the changes of anthropogenic reclamation around the Abandoned Yellow River Delta are divided into the following three stages: 1984 to 1995 is the stage before extensive reclamation, 1996 to 2009 is the extensive reclamation stage, and 2010 to 2019 is the stage after extensive reclamation.

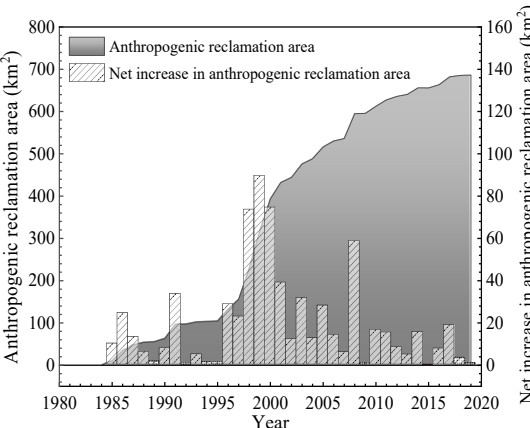

**Figure 6.** Change of anthropogenic reclamation area around the Abandoned Yellow River Delta.

### 4.2. Spatial Distribution of Coastline Migration before Extensive Reclamation

As the intensity of anthropogenic reclamation is relatively low in the early stage, anthropogenic disturbance has a relatively weak influence on the coastline migration, and natural evolution is the main cause of the coastal migration. Therefore, the changes of coastline in the early stage are used to show the spatial distribution characteristics of coastline migration under the natural dynamic condition. As shown in Figure 7, the ordinate represents the coastline migration rate, and the abscissa represents the transect number that gradually increases from 1 to 500 from north to south. The dotted line in the middle is the boundary of the satellite images with different path numbers. Although the coastline migration rate varies greatly from one transect to another, there is still some characteristics. It can be seen obviously that the coastline shows erosion tendency at most of transects in the north part (transect number 1–342) and shows deposition tendency in the south part (transect number 343–500). The transitional area is between Xinyang Port (transect number 333) and Doulong Port (transect number 372). It agrees well with the spatial distribution of the deposition and erosion of the coast reported by Yu and Huang [32].

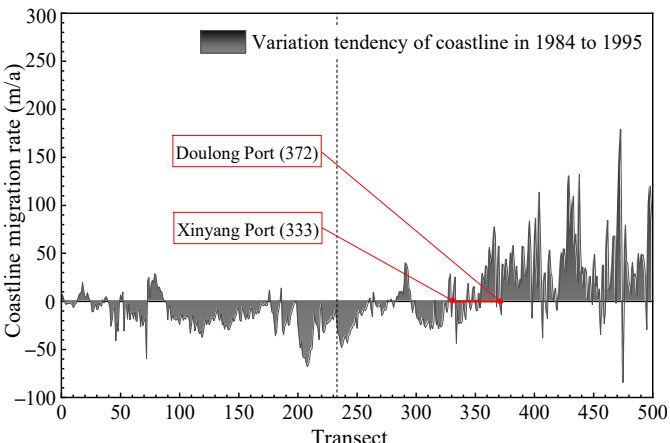

**Figure 7.** Variation tendency of coastline in 1984 to 1995.

Figure 8 shows a comparison of the coastline migration rate with the observed data collected from Yu and Huang [32]. It can be seen that the two reasonably agree with each other. The overall characteristics of the coastal migration obtained from the remote sensing images are consistent with the field survey data. As the survey data are obtained from the measurement of the tidal flat profile at several years during 1980–1997, some differences may occur due to the calculation method of coastline migration in the field survey and present study.

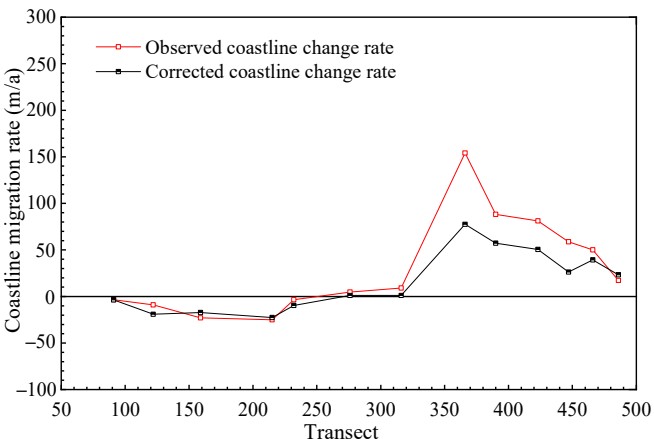

**Figure 8.** Validation of algorithm results.

Local characteristics of coastline migration are summarized in Table 2. Between two river estuaries, the mean coastline migration rate is computed, and the trend is given. Generally, the north part, from Shaoxiang Estuary to Xinyang Port, is more significantly influenced by the diversion of the Ancient Yellow River in 1855. After the sources of sediment transportation were cut off, this part of the coast has undergone persistent erosion. Under the influence of coastal currents, some of the eroded sediment was transported to the south and deposited. Therefore, the southern part, from the Doulong Port to Dongtai Estuary, shows an overall deposition trend. It would be interesting to investigate the response of the natural tidal flat to the extensive reclamation and to see the difference under erosion conditions and deposition conditions. In the following, the concerned coast is divided into the north erosion part and the south deposition part.

**Table 2.** Spatial distribution characteristics of each coast.

| Area | Starting Coast | Ending Coast | Transect | Changes of Coast |
|------|----------------|--------------|----------|------------------|
| North | Shaoxiang Estuary | Sanyu Port | 001–046 | Erosion |
| | Sanyu port | Guan Estuary | 046–074 | Erosion |
| | Guan Estuary | Zhongshan Estuary | 074–132 | Erosion |
| | Zhongshan Estuary | Abandoned Yellow River Estuary | 132–186 | Erosion |
| | Abandoned Yellow River Estuary | Biandan Port | 186–219 | Erosion |
| | Biandan Port | Shuangyang Port | 219–248 | Erosion |
| | Shuangyang port | Sheyang Estuary | 248–290 | Erosion |
| | Sheyang Estuary | Xinyang Port | 290–333 | Erosion |
| | Xinyang Port | Doulong Port | 333–372 | Transition |
| South | Doulong Port | Simaoyou Port | 372–404 | Deposition |
| | Simaoyou Port | Wang Port | 404–438 | Deposition |
| | Wang Port | Chuandong Port | 438–453 | Deposition |
| | Chuandong port | Dongtai Estuary | 453–500 | Deposition |

*4.3. Variation of Tidal Flat Area with Extensive Reclamation*

To show the influence of reclamation, the natural tidal flat area in the north part and the south part at each year has been calculated. The variation tendency of the natural tidal flat area around the Abandoned Yellow River Delta is shown in Figure 9. Before the coastline correction, the changes of the instantaneous tidal flat area with time are scattered; therefore, it is almost impossible to observe the variation tendency of the tidal flat area. After the coastline correction, it can be clearly seen that the tidal flat area basically decreased in both the north part and the south part. It is obvious that the continuous reclamation has greatly reduced the natural tidal flat area. For the north part, the area in 2019 is reduced to only 43% of the original area in 1984. While the reduction in the south part was severer, there is only 27% of the natural tidal flat area left. The decrease in the natural tidal flat area each year greatly corresponds with the reclamation, as shown in Figure 10. In Figure 10, the changes of the anthropogenic reclamation area in the north part and south part are given. Corresponding to the three stages of reclamation, the change of the natural tidal flat area also has three stages, both in the north part and the south part. The change trend shows a relatively slow decrease before an extensive reclamation period, and a rapid decrease during the period, and a slow decrease again in the later period. For the negative increase in anthropogenic reclamation in Figure 10, it is mainly due to the destruction of temporary dikes or the coastal restoration, which has led to a slight increase in the tidal flat area in some years in Figure 9a.

When comparing the north part and the south part, the difference in the variation of the natural tidal flat area can be seen. In the north part, except the years affected by the destruction of temporary dikes, all the years shows a decreasing trend, even in the years with little reclamation. However, it can be seen that in the south part, the natural tidal area can recover after reclamation in some years, for example 1992–1995, after the large reclamation in 1991. We can also see a slight increasing trend in recent years after

the extensive period. It is obvious that the natural tidal flat in the deposition coast can slowly recover, but in the erosion coast it would disappear forever after being occupied by reclamation.

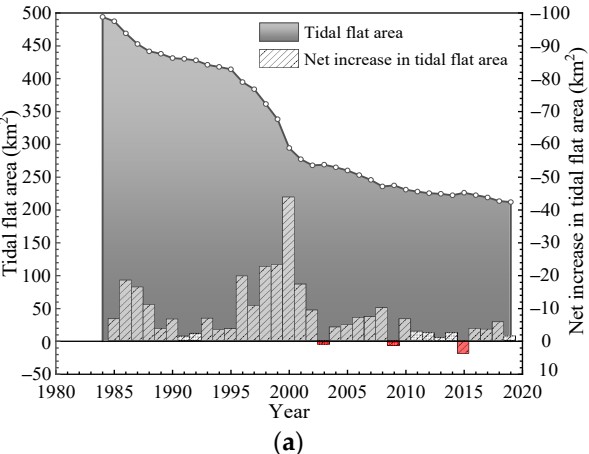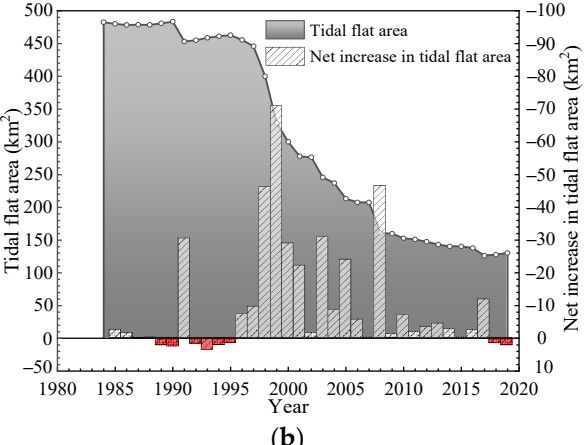

**Figure 9.** (**a**) The variation tendency of natural tidal flat area in the north erosion part; (**b**) The variation tendency of natural tidal flat area in the south deposition part.

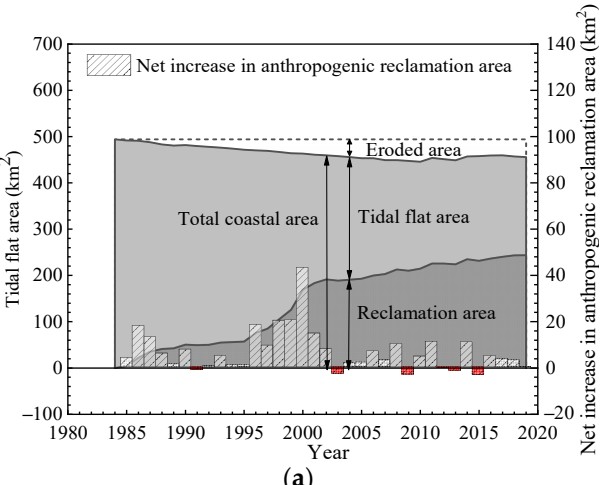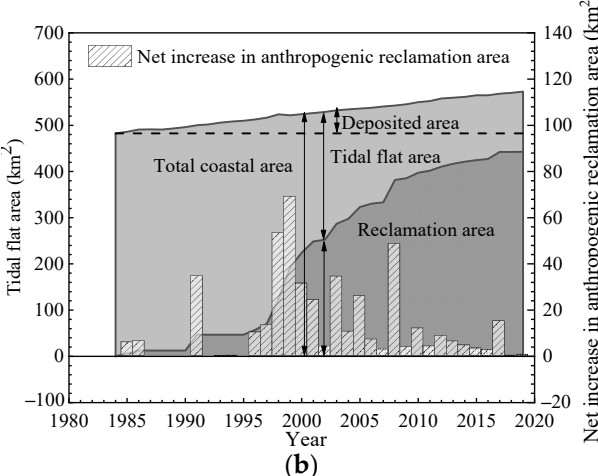

**Figure 10.** (**a**) The changes of anthropogenic reclamation area in the north erosion part; (**b**) The changes of anthropogenic reclamation area in the south deposition part.

Both anthropogenic reclamation and natural evolution have an influence on the variation of the tidal flat, but it is clear that anthropogenic reclamation plays the most important role. The reduction in the natural tidal flat area corresponds to the narrowing of the tidal flat width. Figure 11 shows the tidal flat width at each transect in 1984, 1995, 2010, and 2019. It should be noted that the tidal flat width defined here is the distance of the seawall and datum-base coastline, which is smaller than the width of the intertidal zone.

Whether before, during, or after extensive reclamation, the average width of the tidal flat in the north erosion part is relatively narrower, while relatively wider in the south deposition part. It can be seen that the reduction ratio in the erosion part is smaller than the deposition part. Certainly, the main reason is that reclamation in the deposition transects is much more than that in the erosion transects. However, it is also because of the adjustment of the tidal flat profile. By the coastline correction algorithm, we also can obtain an estimation of the slope of intertidal zone. In the next section, the characteristics of the intertidal zone slope and its response to the extensive reclamation are discussed.

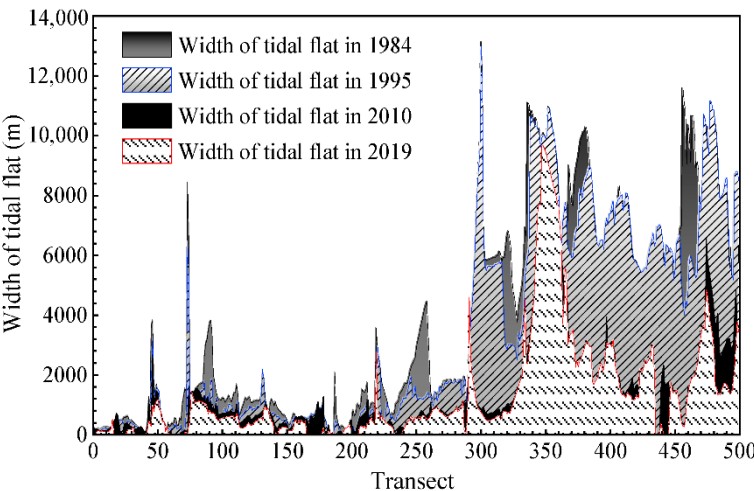

**Figure 11.** The spatial distribution of tidal flat width with time.

### 4.4. Spatiotemporal Changes of Intertidal Zone Slope

In order to analyze the spatial distributions of the intertidal zone slope, a temporal mean slope at each transect is given, as shown in Figure 12. Generally, the slope in the study area is very gentle within the range of 5.5–0.3‰. Although the variation with different transects is large, it can still see as a general feature that the northern erosion coast with a narrow tidal flat has relatively steep slopes, while the southern deposition coast with a wide tidal flat has relatively gentle slopes. The intertidal zone slope from north to south gradually becomes gentler. It should be noted that the slope is calculated along each transect, which may not completely perpendicular to the isobaths.

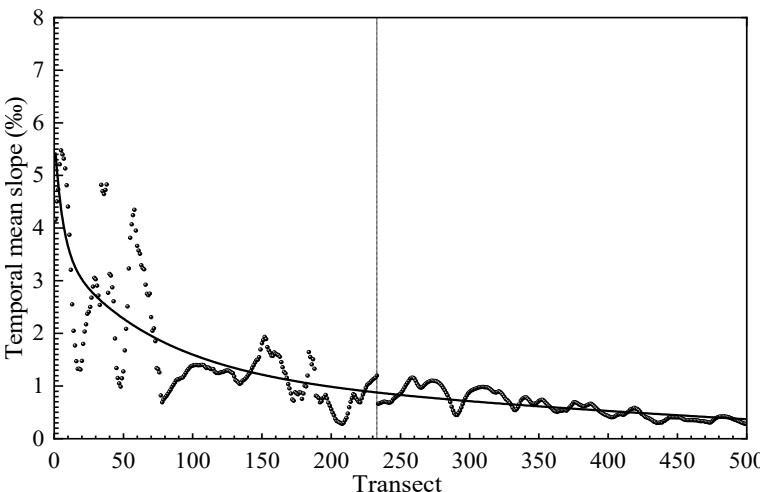

**Figure 12.** The spatial distributions of intertidal zone slope.

As revealed in the previous section, extensive reclamation has been conducted in this area and the width of the tidal flat continues to shrink with time. During this process, the tidal flat profile has also been continuously evolving under natural dynamic conditions. A representative parameter is the intertidal zone slope. In order to see the variation tendency of the intertidal zone slope over time, the averaged intertidal zone slopes in the north part and the south part are calculated, as shown in Figure 13, in order to show the response of the tidal flat profile to the extensive reclamation.

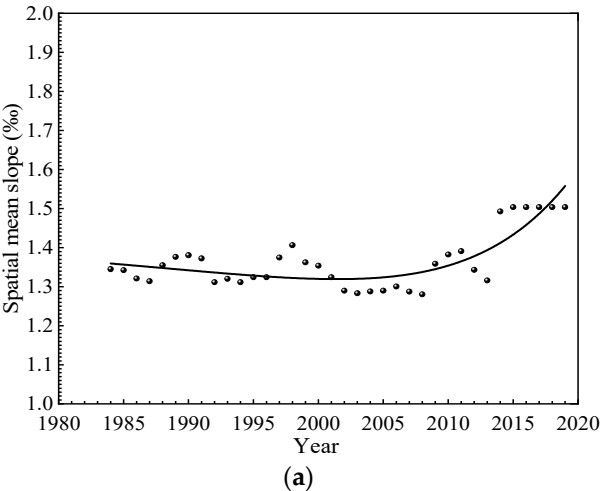 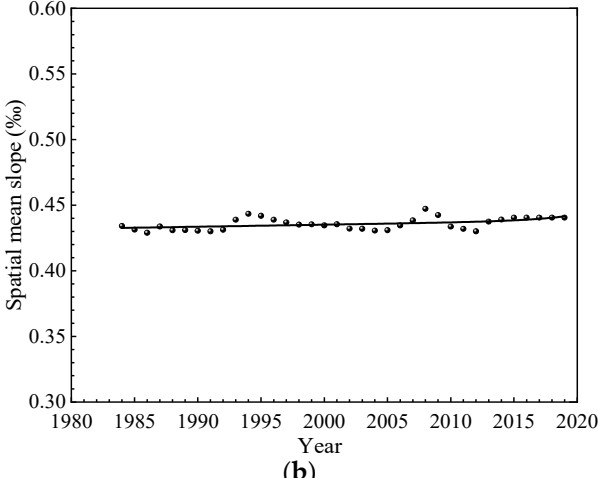

**Figure 13.** (**a**) The temporal distribution of intertidal zone slope in the north erosion part; (**b**) The temporal distribution of intertidal zone slope in the south deposition part.

The difference between the erosion part and the deposition part can be seen. As shown in Figure 13, intertidal zone slopes gradually become steeper and steeper with time in the north erosion part. A clear steepening trend can be seen after the extensive reclamation period. However, in the south deposition part, the overall slope did not change much, although in both parts, the width of the tidal flat gradually becomes narrower. That is because the original tidal flat width is relatively wide in the south part and natural deposition can alleviate the impact of anthropogenic reclamation on the narrowing of tidal flat width to a certain extent. It is further demonstrated that reclamation in deposition coasts has relatively less influence than that in the erosion coast.

### 4.5. Influence of Extensive Reclamation Cumulative Effect on the Biological Habitats

Although extensive reclamation has great impact on the coastline changes, when the reclamation intensities are within a certain range, it will not affect the natural coastline changes. However, it should be noticed that continued reclamation has a cumulative effect. As some species need to survive in a supratidal zone, the disappearance of the supratidal zone may lead to the destruction of biological habitats and have a negative impact on the biodiversity of coastal ecosystems.

In order to explore the year when extensive reclamation has a sharp impact on the changes of the coastline around the Abandoned Yellow River Delta, we counted the number of transects whose supratidal zones disappeared in each year, which is defined in this study as when the datum-based coastline overlaps with the seawall position. The percentage of those transects in the north part and the south part is used to further reveal the influence of extensive reclamation on the coastline changes after 1984, as shown in Figure 14.

It can be seen from Figure 14 that the percentage of coast without a supratidal zone shows an overall increasing trend with time in the north part, among which change trend shows a relatively slow increase before an extensive reclamation period, and a rapid increase during the period, and a slow increase again in the later period. However, in the south deposition part, a sudden change occurs in the year 2010. It is always zero before 2010 and then increases rapidly, which means that the extensive reclamation has sharp influence on coastline changes in 2010.

Both anthropogenic and natural factors in the north erosion part have contributed to the continuous reduction in the tidal flat area, among which extensive reclamation has played a stronger leading role. Although the natural factors in the south deposition part have contributed to the increase in the tidal flat area, the extensive reclamation has also led to the decrease in the tidal flat area. Whether in the north erosion part or the south

deposition part, extensive reclamation is the main reason for the decrease in the tidal flat area, which is also the main driving force for the cumulative effect.

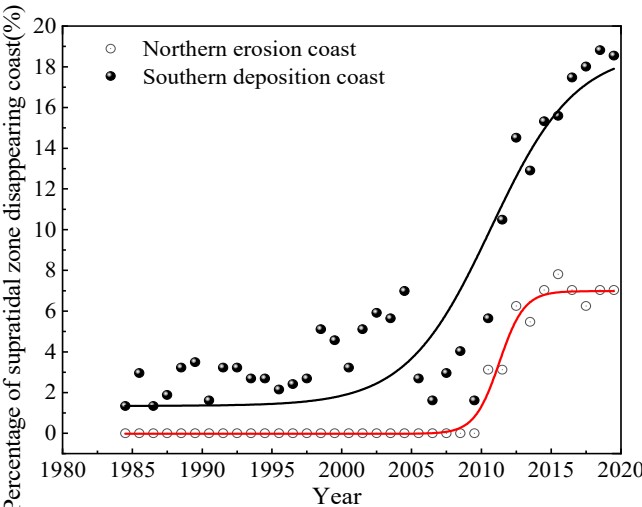

**Figure 14.** Percentage of coast without supratidal zone around the Abandoned Yellow River Delta.

Due to the fact that the width of the tidal flat in the north erosion part is narrow, reclamation has already stimulated the cumulative effect in the early stage, but the cumulative effect is relatively weak, the influence of reclamation on the coastline changes is relatively small; therefore, the percentage of the supratidal zone disappearing coast is relatively low. During the extensive reclamation period, the cumulative effect surged; extensive reclamation has a great impact on the coastline changes, which leads to a further rapid increase in the percentage of the supratidal zone disappearing coast. In the later period, affected by the marine protection policy, the intensity of reclamation becomes lower, the increase in the cumulative effect becomes slow; therefore, the percentage of the supratidal zone disappearing coast slowly increases again.

Although extensive reclamation has led to the reduction in the tidal flat area on the southern deposition coast, due to the fact that the width of tidal flat in the south deposition part is wide and the natural evolution has continuous deposition, reclamation has not stimulated the cumulative effect before an extensive reclamation period and during the period. With the accumulation of the reclamation area in the south deposition part, the cumulative effect was stimulated in 2010; therefore, a sudden change of the percentage of the supratidal zone disappearing coast occurs in the year 2010. However, affected by the marine protection policy in the later period, the cumulative effect in the south part has not been fully increased before it has been flattened. Therefore, the extensive reclamation has greater influence on the coastline changes in the erosion coast with a narrow tidal flat than the deposition coast with a wide tidal flat.

When the extensive reclamation cumulative effect is stimulated, the datum-based coastline of some coasts overlaps with the seawall, leading to the disappearance of the tidal flat habitat in the supratidal zone. With the further accumulation of extensive reclamation, the cumulative effect is gradually obvious, and more coastal biological habitats are lost. Moreover, the study area is an important protected area for migratory bird migration and habitats in the world, and endangered animals such as red-crowned cranes and elk mainly live here; therefore, the destruction of a large area of biological habitats may have a negative impact on the biodiversity of coastal ecosystems. Therefore, appropriate marine protection measures must be strictly strengthened to control the development of the cumulative effect, in order to guide the sustainable development of tidal flat resources around the Abandoned Yellow River Delta in the future.

## 5. Conclusions

In order to analyze the variation tendency of the coastline, tidal flat area, and tidal flat slope under natural and anthropogenic disturbance around the Abandoned Yellow River Delta before, during, and after extensive reclamation during the period of 1984–2019, this study used semi-automatic shoreline extraction technology and a visual interpretation correction method to extract the instantaneous water edges and seawalls from the 500 Landsat series satellite remote sensing images. At the same time, a new coastline correction algorithm had been proposed to obtain the datum-based coastline, aimed to eliminate the influence of varying water level and to better show the variation tendency of the coastline. In addition, the influence of seawalls on the coastlines and the slope estimation were taken into consideration in the algorithm. Based on the obtained coastline and seawall position, the variation tendency of the coastline around the Abandoned Yellow River Delta had been analyzed, comprehensively considering the natural evolution and anthropogenic disturbance.

The results show the variation tendency of the very representative tidal flat coast in recent decades. Affected by the historical diversion of the abandoned Yellow River estuary, this section of the coast shows the characteristics of erosion tendency in the north part and deposition tendency in the south part, which provides a good research object for the comparative analysis of the response of the north erosion coast and the south deposition coast. In addition, this section of the coast has a continuous anthropogenic reclamation from 1984 to 2019, of which the reclamation intensity is the most intense in 1995–2010. Regarding 1995–2010 as the extensive reclamation period, satellite remote sensing images show the changes of the tidal flat area before, during, and after the extensive reclamation period, especially the sharp decrease in the tidal flat area during the extensive reclamation stage. Both anthropogenic and natural factors in the north erosion part have contributed to the continuous reduction in the tidal flat area, among which extensive reclamation has played a stronger leading role and the tidal flat area in 2019 was reduced to only 43% of the original area in 1984. Although the natural factors in the south deposition coast have contributed to the increase in the tidal flat area, the extensive reclamation has led to the decrease in the tidal flat area and the tidal flat area in 2019 was reduced to only 27% of the original area in 1984.

Under the anthropogenic disturbance, we can also see the response of natural evolution to anthropogenic reclamation. Due to the fact that the reclamation area in the south deposition part is larger than the north erosion area, the reduction ratio of tidal flat width in the south deposition part is higher. Actually, the spatial distribution of tidal flat width is relatively narrower in the north erosion part and relatively wider in the south deposition part all the time; even if the reduction ratio of tidal flat width in the south deposition part is higher, the tidal flat width is still relatively wider than the north erosion part. Affected by the extensive reclamation, the adjustment of the tidal flat profile makes the south deposition part have relatively gentler slopes than the north erosion part, and the intertidal zone slopes in the erosion part become steeper along with the extensive reclamation period, while the slope in the deposition part remains stable.

The cumulative effect of anthropogenic reclamation is worthy of attention. The disappearance of the supratidal zone will have influence on the loss of biological habitats and the destruction of coastal ecosystems. The results indicate that there is obvious supratidal zone disappearance during or after the extensive reclamation period in this area. Fortunately, affected by the marine protection policy in the later period, the cumulative effect has been flattened. Therefore, the policy around the Abandoned Yellow River Delta is adjusted in time to avoid further degradation of the intertidal zone, which is benefit to the future protection of biodiversity and the sustainable development of coastal ecosystems.

**Author Contributions:** Conceptualization, Z.S. and X.N.; methodology, Z.S.; validation, Z.S.; resources, Z.S. and X.N.; writing—original draft preparation, Z.S.; writing—review and editing, Z.S. and X.N.; supervision, X.N.; funding acquisition, X.N. Both authors have read and agreed to the published version of the manuscript.

**Funding:** This research was funded by National Key Research and Development Program of China, grant number 2018YFC0407504.

**Data Availability Statement:** All the Landsat data are available at no cost here: https://earthexplorer.usgs.gov/ (accessed on 17 July 2021).

**Acknowledgments:** The authors would like to acknowledge the support by National Key Research and Development Program of China under grant No. 2018YFC0407504. Meanwhile, the authors sincerely thank all anonymous reviewers and the editors for their constructive and excellent reviews that greatly improve the article quality.

**Conflicts of Interest:** The authors declare no conflict of interest.

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
