# Peer review of "Variation Tendency of Coastline under Natural and Anthropogenic Disturbance around the Abandoned Yellow River Delta in 1984–2019"

_remotesensing, doi:10.3390/rs13173391_

Round 1
Reviewer 1 Report
I have reviewed the manuscript by Sun and Niu that talks about the shoreline variation tendency around the abandoned Yellow River Delta. The article uses a semi-automatic satellite-derived shoreline extraction method to analyse the coastline, tidal flat area and slope variation between 1984 and 2019, a period which incorporates extensive reclamation. Whilst I value this contribution, I am inclined to suggest a major revision as the authors need to clarify a few major points, incorporate some important discussions, improve figures and improve several aspects of the text for a broader international community.
One of the key aspects that I think needs attention is related to the validation dataset. The authors state that a dataset extracted from a report written in Chinese was used to validate the accuracy of the satellite-derived results but failed to explain details of how this dataset was acquired, the timespan, the periodicity, etc… This is reflected in the poor analysis conducted in section 4.2 which lacks basic statistics about the applied validation methods. A revised version would benefit from fixing these issues and making this validation dataset available in a public repository.
The authors claim that a new coastline correction algorithm had been proposed following the semi-automated extraction method by Daniels. The advantage of their method is shown in Fig 4, but no discussion is presented in relation to the existing methods published in the international literature. How does the new coastline correction/extraction algorithm compare to existing ones? Improvements can be made to indicate what the differences and benefits are in relation to the works of Luijendijk et al. 2018, Konlechner et al 2010; and Castelle et al. 2021 for instance.
The study area section is extremely short and limited. No meteoceanographic characterisation and contextualisation are provided. What’s the tide amplitude, asymmetry, average wave height, period and direction, winds, longshore transport direction and rates, cyclone frequency, river flow velocities etc?
The international audience would benefit from having all names cited in text in Fig 1. I struggled to find Haihe and Huaihe Rivers for instance and this has made my understanding of the story quite difficult. Besides, the paper would be improved if an insert showing the broad Yellow River and the different stages of abandonment is presented—- labelling all places cited in text including the discharged areas. Fig 10a and 11 also can be improved.
The Abstract needs to be re-written to be more concise and shorter (It currently has ~570 words). Several parts can be taken out to focus on the most important aspects of the article. The same is valid for several parts of the text. Having English as a 2nd language myself, I share the writing difficulties of the authors. While I was able to pick some of the common mistakes below, I am unable to fully address all with the deserved attention. Therefore, I suggest a detail assessment of the English language during revisions to avoid sentences like: “Based on the data of datum-based” (ln556).
Minor:
Ln50 As a transition zone between marine and terrestrial ecosystems, tidal flat contains abundant natural resources. I’m not sure I agree with this opening sentence… Are you suggesting that all coastal landforms contain abundant natural resources? This may be true for tidal flats but not necessarily for others.
Ln61 delete guiding
Ln79 Since the reform and opening up .. which reform are you talking about? What has opened up? Be clear
Ln79 to 86 I suggest a complete rewrite of theses sentences as it was very hard to follow: e.g. the river corse changes and the delta started eroding…how come land reclamation continued? Did the reclamation cause reduction in tidal flat area? What have rare animals to do with this story?
What do you mean by strict protection? Protection of animals, protection of shoreline, please be clear. I think your choice of word is wrong for expose (Ln83)…try better understand and shoreline movement or dynamics…
Ln85 I don’t understand what you mean by this sentence: The coast around the abandoned Yellow River would be as a typical coast with significant changes under anthropogenic disturbance.
I don’t think the AYR coast is typical at all. What’s a typical coast with significant changes? Are you talking about an urban European setting?
Ln197 replace is for being
Ln199 without eliminating
Fig 14 vert axis title-- disappears ?
Ln330 Revise this line to eliminate the show, shown, shown. Put Figs in ().
Luijendijk, A., Hagenaars, G., Ranasinghe, R., Baart, F., Donchyts, G., Aarninkhof, S., 2018. The State of the World's Beaches. Scientific Reports 8, 6641. https://doi.org/10.1038/s41598-018-24630-6
Konlechner, T.M., Kennedy, D.M., O'Grady, J.J., Leach, C., Ranasinghe, R., Carvalho, R.C., Luijendijk, A.P., McInnes, K.L., Ierodiaconou, D., 2020. Mapping spatial variability in shoreline change hotspots from satellite data; a case study in southeast Australia. Estuarine, Coastal and Shelf Science 246, 107018. https://doi.org/10.1016/j.ecss.2020.107018
Castelle, B., Masselink, G., Scott, T., Stokes, C., Konstantinou, A., Marieu, V., Bujan, S., 2021. Satellite-derived shoreline detection at a high-energy meso-macrotidal beach. Geomorphology 383, 107707. https://doi.org/10.1016/j.geomorph.2021.107707
Regards,
Rafael Carvalho
Reviewer 2 Report
The paper is very interesting for the issue, also applicable in other geographical contexts and considering the significant number of satellite images analyzed by the authors. The abstract is perhaps too long compared to the journal's rules and should be shortened. It is not clear how, in the final and partial evaluation, the authors took into account the topography in the considered time-span, considering that it has changed over time due to anthropogenic interference and flooding processes. It would also be useful to insert significant satellite images to show the effects of the change, in addition to the use of histograms, to give greater objective evidence of what is stated in the conclusions and in the data processing.
Reviewer 3 Report
Several parts of your ms are long and not to the point. I propose to reduce the abstract and the introduction by 50% . Also there are a lot of repetitive part in your ms. I propose the conclusions to be numbered.
Round 2
Reviewer 2 Report
The authors have improved the old version and considered the suggestions. They have answered to doubts with clearness. The paper is now publishable in the present form.
Reviewer 3 Report
No more comments the ms is, according my opinion, ready for publicaion.